
# 1 Is dark carbon fixation relevant for oceanic primary
# 2 production estimates?

Federico Baltar[1] and Gerhard J. Herndl[1,2]
[1]Department of Limnology & Bio-Oceanography, University of Vienna, Althanstrasse 14, Vienna,
1090, Austria
[2]NIOZ, Department of Marine Microbiology and Biogeochemistry, Royal Netherlands Institute for Sea
Research, Utrecht University, PO Box 59, 1790 AB Den Burg, The Netherlands
*Correspondence to*: Federico Baltar (federico.baltar@univie.ac.at) and Gerhard J. Herndl
(gerhard.herndl@univie.ac.at)
**Abstract.** About half of the global primary production (PP) is generated in the euphotic layer of the
ocean. The $^{14}$C method developed by Steemann-Nielsen (Nielsen, 1952) more than half a century ago
has been the most frequently used method to determine PP in all aquatic systems. This method includes
dark incubations to exclude the non-phototrophic $CO_2$ fixation. The presence of significant dark DIC
fixation rates has been habitually used to suggest the inaccuracy of the $^{14}$C method to determine
autotrophic phytoplankton primary production. However, we suggest that the dark $CO_2$ fixation rates
should be incorporated into global oceanic carbon production estimates since the total production of
organic matter is not originating only from photosynthesis but also from other processes such as
chemoautotrophic and anaplerotic processes. Here we analyzed data collected over almost 30 years
from the longest available oceanic time series and calculated that the inclusion of dark dissolved
inorganic carbon (DIC) fixation would increase oceanic PP estimates by 5-22% when total dark DIC
fixation is included or by 2.5-11% when only considering the nighttime DIC fixation. We conclude that
dark DIC fixation should be included into global oceanic primary production estimates as it represents
newly synthesized organic carbon (ca. 1.2 -11 Pg C y$^{-1}$) available for the marine food web.

## 28 1 Introduction

Primary production (PP) is arguably one of the most important metabolic processes, and half of the
global PP is generated in the euphotic layer of the ocean (Field et al., 1998). Thus, it is crucial to
accurately estimate marine PP rates. The $^{14}$C method to estimate aquatic primary production is based on
incubating environmental water samples with a known concentration of $^{14}$C-bicarbonate, and measure
the concentration of $^{14}$C incorporated into microbial biomass, i.e, the conversion rate of inorganic to
organic carbon. One of the key issues associated with the interpretation of the results derived from this
method is the need to assume that dissolved inorganic carbon (DIC) uptake is associated essentially
only with photosynthetic activity of phytoplankton (Harris et al., 1989; Ignatiades et al., 1987;
Legendre et al., 1983; Petersen, 1979; Prakash et al., 1991; Taguchi, 1983). That implies that dark DIC
fixation of other organisms like heterotrophs or chemoautotrophs is considered insignificant, because if
substantial DIC fixation would occur in the dark then this method would not be a reliable measure of
photosynthetic primary production (Prakash et al., 1991). Although Steeman Nielsen originally thought



that dark fixation rates would only amount to about 1% of DIC fixation in the presence of solar
radiation, he promptly realized that dark DIC fixation could be up to >50% of that under solar radiation
(Nielsen, 1960; Prakash et al., 1991). Despite these findings, the standard protocol of the [14]C method,
analyses and interpretation of the data have remained essentially unchanged for decades.
However, over the past two-three decades our understanding of the metabolic potential of marine
microbes has expanded dramatically. It is now accepted that, besides autotrophic phytoplankton, there
are many chemoautotrophs and hetero- and mixotrophs inhabiting the oxygenated upper ocean with the
ability to mediate dark DIC fixation. A great metabolic potential related to DIC fixation was uncovered
with the development and application of (meta)genomic tools to marine microbial communities
(Moran, 2008). High dark DIC fixation rates attributed to chemoautotrophic and heterotrophic
prokaryotes have been reported in surface (Alonso-Sáez et al., 2010; Li and Dickie, 1991; Li et al.,
1993; Markager, 1998; Prakash et al., 1991), and the deep ocean (Baltar et al., 2010; Baltar et al., 2016;
Herndl et al., 2005; Reinthaler et al., 2010). In particular, the rates of DIC fixation parallel those of
prokaryotic heterotrophic production in the deep ocean (Baltar et al., 2016; Reinthaler et al., 2010). The
contribution of the organic carbon supplied by dark DIC fixation to the prokaryotic carbon demand in
the deep ocean is comparable to the supply of sinking particulate organic carbon flux (Baltar et al.,
2010). DIC fixation due to chemoautotrophy is assumed to be relatively more important in aphotic than
photic waters due to the reported light sensitivity of ammonia oxidation which is a chemoautotrophic
process (citation on light sensitivity). However, substantial chemoautotrophy such as nitrification was
found to take place not only in the meso- but also in epipelagic waters, where it plays a significant role
in providing N for oceanic new production (Yool et al., 2007). In general, chemoautotrophy is
widespread in the marine environment amounting to an estimated global oceanic DIC fixation of 0.77
Pg C per year (Middelburg, 2011). This estimated DIC fixation rate is similar to the amount of organic
C supplied by the worlds' rivers and buried in oceanic sediments (Middelburg, 2011).
DIC fixation is not only performed by photoautotrophs, but chemoautotrophs and heterotrophs
incorporate $CO_2$ via a wide range of carboxylation reactions (anaplerotic reactions and the synthesis of
fatty acids, nucleotides and amino acids) that form part of their central and peripheral metabolic
pathways (Dijkhuizen and Harder, 1984; Erb, 2011). Since many ecologically relevant compounds are
metabolized via these "assimilatory carboxylases", it has been recently suggested that these enzymes
can be relevant for the global C cycle along with "autotrophic carboxylases" (Erb, 2011). In the ocean
in particular, anaplerotic DIC incorporation plays an important role in compensating metabolic
imbalances in marine bacteria under oligotrophic conditions, contributing up to >30% of the carbon
incorporated into biomass (González et al., 2008; Palovaara et al., 2014). Moreover, it has also been
shown that if the heterotrophic metabolism of bacteria is suddenly intensified (e.g., after an input of
organic matter), dark DIC fixation rates and the expression of transcripts associated to key anaplerotic
enzymes increase proportionally (Baltar et al., 2016). Considering the oligotrophic nature of most of
the ocean and the sporadic, pulsed input of organic matter it is possible that anaplerotic reactions may
at times contribute a larger proportion to dark (and total) DIC fixation. However, despite evidence of



dark DIC fixation taking place, it remains unknown how much anaplerotic reactions contribute to
oceanic DIC fixation.
Bearing all these discoveries on oceanic DIC fixation in mind, it is not surprising that the dark DIC
fixation rates have been an issue for the interpretation of the [14]C method to measure phytoplankton PP.
Traditionally, the way to deal with the dark fixation in the [14]C method is to perform light and dark
incubations, and subtract the rates obtained under dark conditions from that in the light incubations.
The presence of significant dark DIC fixation rates has been habitually attributed to the inaccuracy of
the [14]C method to determine phytoplankton primary production.
However, we believe that it might be sensible to go a step further and suggest that the dark DIC
fixation rates measured with the [14]C method should be incorporated into global carbon production
estimates. In the oceanic environment, the total production of organic matter is not only originating
from photosynthesis but also from chemoautotrophic and anaplerotic processes. These other DIC
fixation pathways also produce organic C not only in the daytime but also during nighttime. Thus,
although it makes sense to exclude the dark DIC fixation rates if the aim is to estimate
photoautotrophic production only, dark DIC fixation (at least the one occurring during the nighttime)
should actually be added to the photoautotrophic production if we want to arrive at a realistic estimate
on total organic carbon production via DIC fixation.

## 2 Contribution of dark inorganic carbon fixation to overall oceanic photoautotrophic carbon dioxide fixation

Here, we used the publically available data on the [14]C PP method from the longest oceanic time series
stations (ALOHA [22°45′N 158°00′W] and BATS [31°40′N 64°10′W]) to determine the relative
importance of dark DIC fixation relative to light-based DIC fixation in the epipelagic ocean. Herein, PP
refers to the traditional way of estimating PP in the ocean (i.e., the carbon fixed in the light minus that
fixed in the dark incubation). We defined "total DIC fixation" as the sum of light + dark DIC fixation.
First we compared the temporal and vertical changes in the ratio between dark and light DIC fixation.
Then, we integrated the rates and used the stoichiometry of nitrification to calculate the overall relative
contribution of dark DIC fixation and nitrification-based DIC fixation to the dark and total organic
carbon production. With this, we aim at providing an estimate on the amount of C being missed with
the traditionally light-based PP estimates, and make a case for the inclusion of the dark DIC fixation in
oceanic organic carbon production estimates.
The available data (i.e., light and dark DIC fixation rates) were obtained from the databases of BATS
between 1989 and 2017 and of ALOHA between 1989 and 2000 (Fig. 1). The maximum sampling
depth was deeper for ALOHA (175 m) than for BATS (150 m). Yet, both the ALOHA and BATS
station showed a pronounced increase with depth in the dark to light DIC fixation ratio spanning from 0
to >2.5 (Fig. 1). This ratio of dark to light DIC fixation was generally lower at ALOHA than at BATS,
particularly in the top 100 m layer. A clearer and stronger seasonality was found for BATS than for
ALOHA, provoked by differences in stratification during the summer and vertical mixing during the
winter due to their differences in latitude (Fig. 1 and 2). Interestingly, in the BATS dataset, there was a
tendency detectable towards a higher ratio of dark to light DIC fixation in the top half of the euphotic
layer (0-65 m) from the year 2012 to 2017 than in the preceding years. It is not clear what the reason
might be for this increase in the dark to light DIC fixation ratio in recent years. It might be associated,
however, to changes in the vertical structure of the water column over this time span as indicated in the
shifts observed in temperature, salinity and sigma-t during the same period. The $\sigma_t$ isopycnal of 26
reached and remained deeper than 200 m during the years 2012-2017 (Fig. 2). This has caused a
deepening of the mixed layer, causing a decrease in chlorophyll-*a* concentrations in shallow waters and
a deepening of the deep chlorophyll maximum (Fig. 2D).
We then compiled and integrated the data for all available depths (down to 150 and 175 m at BATS
and ALOHA, respectively) to calculate how much the inclusion of dark DIC fixation would increase
the total PP estimates in the epipelagic waters (Table 1). Due to the strong vertical differences observed
in the ratio of dark to light DIC fixation (Fig. 1), we also decided to subdivide the integration of the
epipelagic water column into a shallow and a deep layer. At ALOHA, the inclusion of dark fixation
would increase PP by 3.7% in the shallow layer (0-65 m) and by 8.6% in the deep layer (65-175 m).
When integrating for the whole depth range of the euphotic layer at ALOHA, the inclusion of dark
fixation increases PP estimates by 5.1%. At BATS, this contribution is much higher with 17.3% and
36.5% for the shallow (0-70 m) and deep (70-150 m) layer. When integrated for the whole water
column, the dark DIC fixation increases PP estimated at BATS by 22.1%.
To estimate the potential relative contribution of chemoautotrophy and anaplerotic reactions to dark
DIC fixation, we calculated the potential proportion of nitrification to dark DIC fixation based on the
global euphotic nitrification rate of 0.195 $d^{-1}$ obtained by (Yool et al., 2007). For that we used
published $NH_4^+$ concentrations from ALOHA (Segura-Noguera MM et al., 2014) and from BATS
(Lipschultz, 2001). The calculated depth-integrated ammonium oxidation by this method (1.5 mmol $m^{-2}$
$d^{-1}$) is remarkably similar to the rate (1.6 mmol $m^{-2}$ $d^{-1}$) obtained by Dore & Karl (Dore and Karl,
1996) for ALOHA using inhibitor-sensitive dark $^{14}C$ uptake assays. We then used the stoichiometry of
ammonia oxidation (i.e., ratio of $CO_2$ fixed per $NH_4^+$ oxidized of 0.1) to calculate the potential
contribution of ammonia oxidation (nitrification) to the dark DIC fixation. The remaining dark fixation
was assumed to originate from other chemoautotrophic processes and anaplerotic metabolism. We
found that the integrated contribution of nitrification to dark DIC fixation is relatively low at both
stations (8.8% and 2% at ALOHA and BATS, respectively), suggesting that most of the dark fixation
(91.2 and 98% at ALOHA and BATS, respectively) is performed by chemoautotrophs other than
ammonia-oxidizers and/or anaplerotic metabolism.
Since C fixation occurs both at daytime (photosynthesis, chemosynthesis, anaperotism) and nighttime
(chemosynthesis, anaplerotism), a more appropriate measure of the total PP would include the DIC





fixation over the entire day (and not only during daytime). The DIC fixation in the light incubation
represents the fixation performed by all organisms (photoautotrophs, chemoautotrophs and anaplerotic
metabolism) hence, including dark fixation during the daytime. The DIC fixation in the dark bottle
accounts for the DIC fixation by all organisms during the nighttime. Assuming that the dark DIC
fixation is constant during over the diel cycle, we can calculate the nighttime DIC fixation by dividing
the dark daily DIC fixation (in mg C m$^2$ d$^1$) by half (assuming a 12 h dark period). That would imply
that the inclusion of dark DIC fixation in PP estimates would increase total PP (DIC fixation) by 2.5%
at ALOHA and 11% at BATS.  It is important to realize that for anaplerotic DIC fixation this would be
a conservative estimate since it has been observed that proteorhodopsin-harboring heterotrophic marine
bacteria increase their DIC fixation due to anaplerotic reactions in response to light (González et al.,
2008; Palovaara et al., 2014). Moreover, chemoautotrophic DIC fixation rates such as nitrification are
reduced in the presence of light. Thus, the chemoautotrophic fixation taking place in the light bottles
also represent a conservative estimate.

**3 Conclusions and implications**
Collectively, these results suggest that including total dark DIC fixation into actual PP estimates
increases the total PP rates by 5 and 22% at ALOHA and BATS, respectively, and by 2.5 to 11% when
only the nighttime DIC fixation is considered. Considering a net primary production
(photoautotrophic) in the global ocean (Field et al., 1998) of ca. 50 Pg C y$^{-1}$, this range of contribution
of the dark DIC fixation (2.5 to 22% of total PP) would translate into ca. 1.2 to 11 Pg C y$^{-1}$. To put
these numbers into context, the C flux associated to dark ocean (>200 m) chemoautotrophy is 0.11 Pg
C y$^{-1}$, and the total respiration C fluxes in the global ocean sediments, the dark ocean and in the
euphotic zone are 1.2, 7.3 and 44 Pg C y$^{-1}$, respectively (Dunne et al., 2007; Middelburg, 2011). This is
a substantial amount of organic C produced via DIC fixation currently not accounted for in global C
budget estimates, which might have implications for the carbon cycling by the heterotrophic food web.
For instance, this, thus far, largely ignored and thus unaccounted source of newly synthesized organic
C might help resolving the contrasting views of whether the ocean is net heterotrophic or net
autotrophic (Duarte et al., 2013; Ducklow and Doney, 2013; Williams et al., 2013), as well as reconcile
the imbalance between the deep ocean heterotrophic C demand and the sinking particulate organic C
flux (Baltar et al., 2009; Burd et al., 2010; Steinberg et al., 2008). Moreover, the relevance of
incorporating this dark DIC fixation in the estimates of total PP might become even more crucial if the
tendency continues towards an increasing ratio of dark to total PP we observed over the past five year
period for BATS. Overall, we suggest that the DIC fixation measured with the $^{14}$C method under dark
conditions (particularly during nighttime) should be seen as an integral part of the global ocean PP
generating new particulate organic carbon potentially available for the marine food web.



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

**Acknowledgments**
We would like to acknowledge the great effort of BATS (Bermuda Atlantic Time-series) and ALOHA
(A Long-term Oligotrophic Habitat Assessment) stations for generating and making publically
available their data. This study was funded by the Austrian Science Fund (FWF) project ARTEMIS
(P28781-B21) to GJH, and the Rutherford Discovery Fellowship (by the Royal Society of New
Zealand).


**Authors contribution**
F.B. and G.J.H contributed equally to the development of the paper.


**Data availability statement**
All data are available and were downloaded from the BATS (Bermuda Atlantic Time-series) and
ALOHA (A Long-term Oligotrophic Habitat Assessment) stations websites.


**Competing interests**
The authors declare no competing interests.






**Table 1.** Integrated total primary production (PP) (i.e., light – dark DIC fixation), dark DIC fixation and percentage of dark to total PP at station ALOHA (0-175 m) from 1989 to 2000 (11 y) and at station BATS (0-150 m) from 1989 to 2017 (29 y). The contribution of nitrification to dark fixation was calculated based on the global euphotic nitrification rate of 0.195 d$^{-1}$ (Yool et al., 2007) using published $NH_4^+$ concentrations from ALOHA (Segura-Noguera et al., 2014) and from BATS

(Lipschultz 2001). The stoichiometry of ammonia oxidation (ratio of $CO_2$ fixed per $NH_4^+$ oxidized of 0.1) was used to calculate the potential contribution of ammonia oxidation (nitrification) to the dark $CO_2$ fixation. The remaining dark fixation was assumed to be from other chemoautotrophic and anaplerotic processes.

| ALOHA | | | | |
|---|---|---|---|---|
| Depth range (m) | Total PP (mg C m$^{-2}$ d$^{-1}$) | Dark DIC fixation (mg C m$^{-2}$ d$^{-1}$) | % of dark to total PP | % of dark to total PP (calculated for daily 12h dark fix) |
| 0-65 | 289.1 | 10.7 | **3.7** | **1.8** |
| 65-175 | 117.5 | 10.1 | **8.6** | **4.3** |
| 0-175 | 406.6 | 20.8 | **5.1** | **2.5** |

| Depth range (m) | nitrification (mmol $NH_4^+$ m$^{-2}$ d$^{-1}$) | % dark DIC fixation from nitrification | % dark DIC fixation from other chemolitho-autotrophic and anaplerotic reactions | % of dark DIC fixation from other chemolithoautotrophic and anaplerotic processes to total PP |
|---|---|---|---|---|
| 0-70 | 0.5 | 5.4 | 94.6 | 3.5 |
| 70-150 | 1.1 | 12.5 | 87.5 | 7.5 |
| 0-150 | 1.5 | 8.8 | 91.2 | 4.7 |





| BATS | | | | |
|---|---|---|---|---|
| Depth range (m) | Total PP (mg C m⁻² d⁻¹) | Dark DIC fixation (mg C m⁻² d⁻¹) | % of dark to total PP | % of dark to total PP (calculated for daily 12h dark fix) |
| 0-70 | 314.2 | 54.3 | **17.3** | **8.6** |
| 70-150 | 103.8 | 37.9 | **36.5** | **18.2** |
| 0-150 | 418.0 | 92.2 | **22.1** | **11** |

| Depth range (m) | nitrification (mmol NH₄⁺ m⁻² d⁻¹) | % of dark DIC fixation from nitrification | % of dark DIC fixation from other chemolithoautotrophic and anaplerotic processes | % of dark DIC fixation from other chemolithoautotrophic and anaplerotic processes  to total PP |
|---|---|---|---|---|
| 0-70 | 0.7 | 1.5 | 98.5 | 17.0 |
| 70-150 | 0.9 | 2.7 | 97.3 | 35.5 |
| 0-150 | 1.6 | 2.0 | 98.0 | 21.6 |





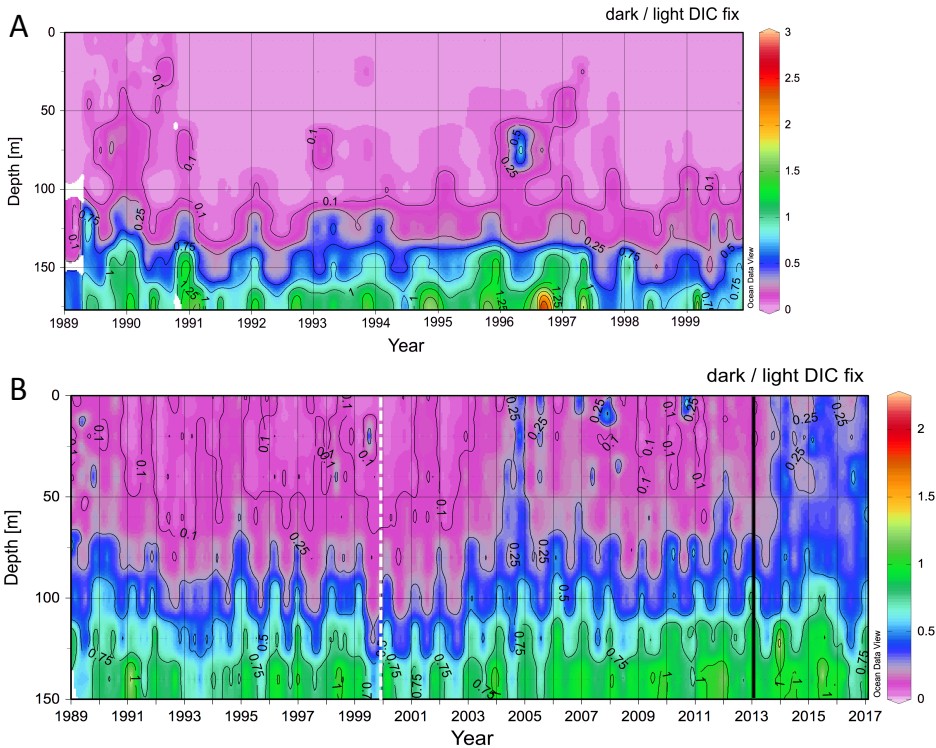

**Fig 1**. Variation in the ratio of dark to light DIC fixation rates (A) at ALOHA (from 1989 to 2000) and (B) at BATS (from
1989 to 2017). The dashed line in the plots for BATS indicates the recent years in record in the ALOHA dataset. The solid
black line highlights a potential shift in the year 2013.





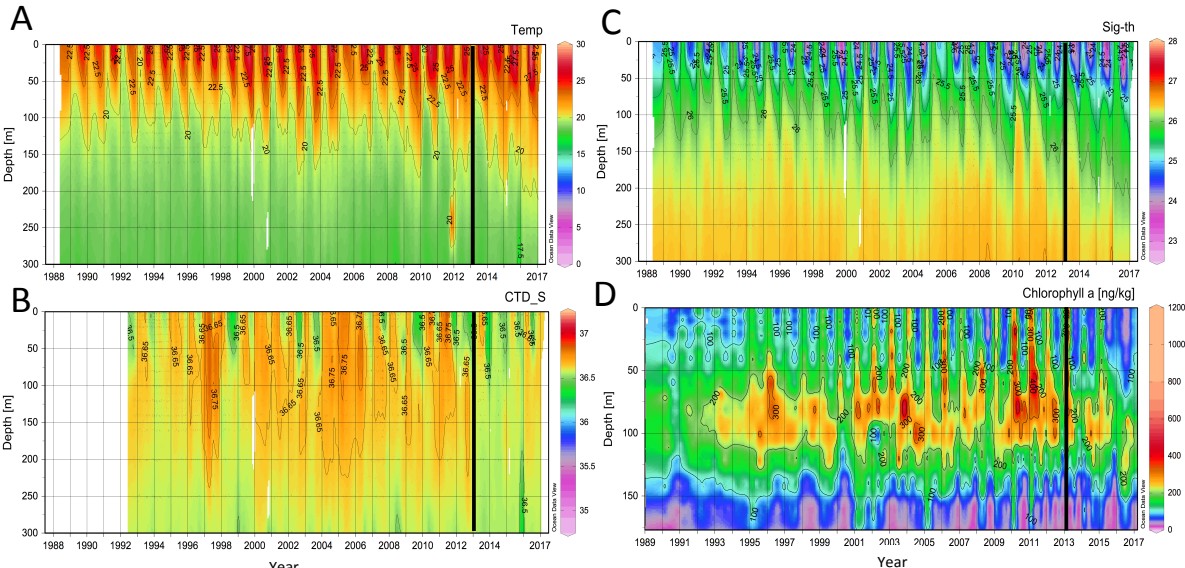

**Fig 2**. Variation in (A) temperature (°C), (B) salinity, (C) sigma-t, and (D) Chlorophyll-*a* at BATS (from 1989 to 2017). The solid black line highlights a potential shift in the year 2013.
