# Peer review of "Is dark carbon fixation relevant for oceanic primary production estimates?"

_Biogeosciences, 2019_

## Short Comment (SC1) · 18 Jun 2019

One very brief comment on an angle that doesn't appear to be mentioned in the manuscript at present, and which might be relevant for biogeochemical budgeting.

While the "dark fixation" of carbon via nitrification is not directly fed by solar energy, it relies on the availability of a substrate (ammonia / ammonium) that itself is a breakdown product of organic molecules that were originally fashioned using solar energy. If, instead of carbon, biogeochemical transformations are viewed through the prism of energy, this dark fixation is part of the dissipation / remineralisation loop rather than a truly new source of biomass.

In particular places, or at particular times, it may look like dark fixation is providing a quantitatively significant alternative to the "mainstream" photoautotrophic pathways, but ultimately it is really spun off the back of earlier (and potentially unobserved) sunlight-driven primary production. A clear exception is around systems such as hydrothermal vents, where the chemical substrates driving dark fixation do not have a solar source, but these are a minor component at the global scale.

This is not, of course, to dismiss dark fixation. It is an important process (or series of processes), the understanding of which is critical if we are to fully understand biogeochemical cycles in the ocean. Particularly so, as here, where the details of measurement require a proper accounting of its operation. But in tracing the details of these cycles, it is important to still recognise the underlying drivers, even if these are indirectly coupled through long chains of chemical transformation.

---

## Referee Comment (RC1) · Anonymous Referee #1 · 19 Jul 2019

General comments

The manuscript mentioned above by Baltar & Herndl estimates the importance of dark carbon fixation (e.g. chemoautotrophic production plus anaplerotic reactions) in carbon budgets in the ocean. Authors use a novel approach to interpret routinely measured data, such as dark bottle incubations used for primary production estimations, to calculate dark carbon fixation rates. Interestingly the two datasets studied show a) that dark carbon fixation estimates are equal to 2.5 -22% of the phototrophic carbon fixation, b) there is a seasonal effect on the ratio of dark and light carbon fixation, especially for one site and, c) nitrification accounts for only a minor proportion (2-9%) of the total

dark carbon fixation. These results are of relevance because they clearly show that dark carbon fixation in the euphotic layer can substantially increase the PP estimates. This estimated dark carbon fixation corresponds to a production of 1.2 to 11 Pg C y-1 which scales well with the carbon respiration fluxes reported for ocean sediments and the dark ocean. Thus dark carbon fixation rates should be considered in future carbon budget studies of the ocean. Furthermore, the fact that nitrification plays a minor role in carbon cycling in the euphotic ocean suggests that the major sources of dark carbon fixation are unknown and deserve further study. In general, I find the manuscript well prepared and related work is well credited. To my knowledge, the methodology used is sound and I find no reason to doubt the interpretations of the data. I therefore recommend publication of the manuscript with minor corrections.

Specific comment

The results indicate that nitrification explains less than 10% of the total dark production. The Authors thus state that other chemoautotrophic processes (different from nitrification) and heterotrophic processes (anaplerotic reactions) should account for the remaining 90%. Assuming anaplerotic reactions account for 30% of this dark carbon fixation (line 72), then more than half of the activity remains un-assigned to a specific process. I ask the Authors to briefly discuss or present a specific hypothesis as to which other autotrophic processes may be involved, or suggest methods that can be used to unravel the sources of this unknown dark carbon fixation activity.

Technical corrections

Abstract: Choose one term throughout the text for consistency: "dark carbon fixation", "dark DIC fixation" or "dark CO2 fixation".

Line 59: text states "citation on light sensitivity" please include reference

Line 114: please state actual maximum value instead of ">2.5"

Line 118" tendency detectable towards" change wording

line 122: replace "sigma-t" with "density ($\sigma$t)" ; after "[...] same period" add "Fig. 2)."

Line 123: change "(Fig 2)" to "(Fig 2C)".

Line 123-125: When explaining the seasonality observed at Bats for the dark to light carbon fixation ratio the Authors mention a deepening of the deep chlorophyll maximum but do not explicitly describe how this affects the calculated ratio. I ask the Authors to rephrase these sentences so that the readers can clearly understand the connexion that is currently implied in the text.

Line 136-149: include a reference to Table 1 to guide the reader through the calculations

Line 138: change "[. . .] by (Yool et al., 2007)" for "[. . .] by Yool et al. (2007)"

Line 139-140: state the NH4+ values taken from Segura-Noguera MM et al 2014 and Lipschultz 2001 to estimate nitrification either in the text or in table 1.

Line 141: Dore & Karl are written twice. Correct reference

Line 156: "during over the diel cycle" please correct wording

Line 158-159: add a reference to Table 1

Line 163: add citation for the concept that nitrification is reduced in light

Figure 1: keep the same range for the ratio for both plots, from 0 to 3, so that the colour scheme is the same.

Figure 2: remove names from plots (upper right corner) or write correctly (not Temp but Temperature)

Table 1: last column "% of dark DIC fixation from other chemolithoautotrophic and anaplerotic processes to total PP" is not explained or referred to in the text for either site. Please erase from table. Change "chemolithoautotrophy" for "chemoautotrophy"

[Figure]

---

## Referee Comment (RC2) · Anonymous Referee #2 · 29 Jul 2019

This is an interesting small paper that reviews data on dark 14C incorporation in the ocean and that postulates that it amounts to a relevant % of total primary production and that should be considered in evaluations of global primary production. I'm sympathetic with the author's effort as I had somehow surprisingly been puzzled by the lack of reference to dark C fixation (which it was a classic in the 80s, considered as "errors" of the Steeman-Nielsen method) I like the paper, I find the issue sensitive, and the analysis is certainly worthwhile. There are only a couple of points that could be discussed and that would benefit the ms. First point is stated at line 85: "dark C fixation had been attributed to the inaccuracy of the 14C method. . ." Could you expand on that? Could you tell the reader why the authors at the time thought this was an error? Why dark

fixation was never considered primary production? Maybe this was due to the authors considering dark fixation as, at least in part, abiotic fixation? How do you deal with abiotic fixation in your estimates? A second point concerns to the night extrapolation of the daytime dark incorporation rates. The authors correctly identify mechanisms by which one should not assume nighttime fixation to be equal to daytime fixation (lines >160). However, I wonder how diel changes in organism activity or in water chemistry warrant that the daytime dark fixation should be above or below the night time value. Did anyone ever measure nighttime dark fixation? A third issue that could be expanded is the Table 1 increase in % dark incorporation in the 70-150 m layer. I think it was a good idea splitting the calculations by layer, but you should maybe make very clear whether this layer contains the DCM in all cases and then speculate as to why the DCM or the layer below the DCM should have a larger proportion of chemoautotrophs or anaplerotic reactions. Also, maybe the layer split could be made more clearly separating above-DCM, in-DCM and below-DCM depths. Finally, I'm uneasy about the 4x difference in estimations between ALOHA and BATS. I can't find any hint of the reasons for the differences, other than different people doing the estimations. You should recognize this difference and suggest an explanation if at all possible. Can the differential oceanography of both sites play a role? Also, and about the shift of dark C fixation (or at least the proportion) occurring at BATS after 2013, I would appreciate a little bit of hypothesis-building providing a mechanistic linkage between the deepening of the mixed layer and the beneficial? effect on anaplerotic fixation (why should it be benefited?) or chemoautotrophy. . . And just a tiny other comment: l. 59. Citation missing here!

Good paper that should be published. My comments point to clarifications and further insight that would, I believe, make the authors' point even stronger.

---

## Referee Comment (RC3) · Andrew K. Sweetman (Referee) · 16 Aug 2019

Having started working on this topic in deep-sea sediments myself, I found the manuscript very interesting, well written and worthy of publication in Biogeosciences. I have a few comments and suggested changes that I think would be good to incorporate before being finally accepted in Biogeosciences. I wish the authors all the best in revising the paper and look forward to the final publication. Please find my suggestions and comments in the attached PDF. Sincerely, Andrew K. Sweetman

Please also note the supplement to this comment:

[Figure]

https://www.biogeosciences-discuss.net/bg-2019-223/bg-2019-223-RC3-supplement.pdf

---

## Author Comment (AC1) · 3 Sep 2019

Andrew Yool axy@noc.ac.uk

One very brief comment on an angle that doesn't appear to be mentioned in the manuscript at present, and which might be relevant for biogeochemical budgeting. While the "dark fixation" of carbon via nitrification is not directly fed by solar energy, it relies on the availability of a substrate (ammonia / ammonium) that itself is a break-down product of organic molecules that were originally fashioned using solar energy. If, instead of carbon, biogeochemical transformations are viewed through the prism of energy, this dark fixation is part of the dissipation / remineralisation loop rather than

a truly new source of biomass. In particular places, or at particular times, it may look like dark fixation is providing a quantitatively significant alternative to the "mainstream" photoautotrophic path- ways, but ultimately it is really spun off the back of earlier (and potentially unobserved) sunlight-driven primary production. A clear exception is around systems such as hydrothermal vents, where the chemical substrates driving dark fixation do not have a solar source, but these are a minor component at the global scale. This is not, of course, to dismiss dark fixation. It is an important process (or series of processes), the understanding of which is critical if we are to fully understand biogeochemical cycles in the ocean. Particularly so, as here, where the details of measurement require a proper accounting of its operation. But in tracing the details of these cycles, it is important to still recognise the underlying drivers, even if these are indirectly coupled through long chains of chemical transformation.

Response: Thank you for your comment. We now mentioned this aspect in the manuscript (p.2, l.65-67); it reads: "Yet, while the dark DIC fixation via nitrification is not directly fed by solar energy, it relies on the availability of a substrate (ammonia / ammonium) that itself is a break-down product of organic molecules that were originally fashioned using solar energy."

---

## Author Comment (AC2) · 3 Sep 2019

**Reviewer #1**

1. Reviewer General comments The manuscript mentioned above by Baltar & Herndl estimates the importance of dark carbon fixation (e.g. chemoautotrophic production plus anaplerotic reactions) in carbon budgets in the ocean. Authors use a novel approach to interpret routinely measured data, such as dark bottle incubations used for primary production estimations, to calcu- late dark carbon fixation rates. Interestingly the two datasets studied show a) that dark carbon fixation estimates are equal to 2.5 - 22% of the phototrophic carbon fixation, b) there is a seasonal effect on the ratio of dark

and light carbon fixation, especially for one site and, c) nitrification accounts for only a minor proportion (2-9%) of the total dark carbon fixation. These results are of relevance because they clearly show that dark carbon fixation in the euphotic layer can substantially increase the PP estimates. This estimated dark carbon fixation corresponds to a production of 1.2 to 11 Pg C y-1 which scales well with the carbon respiration fluxes reported for ocean sediments and the dark ocean. Thus dark carbon fixation rates should be considered in future carbon budget studies of the ocean. Furthermore, the fact that nitrification plays a minor role in carbon cycling in the euphotic ocean suggests that the major sources of dark car- bon fixation are unknown and deserve further study. In general, I find the manuscript well prepared and related work is well credited. To my knowledge, the methodology used is sound and I find no reason to doubt the interpretations of the data. I therefore recommend publication of the manuscript with minor corrections.

Comment: We honestly appreciate the positive comments and the support of the reviewer.

2. Reviewer Specific comment The results indicate that nitrification explains less than 10% of the total dark produc- tion. The Authors thus state that other chemoautotrophic processes (different from ni- trification) and heterotrophic processes (anaplerotic reactions) should account for the remaining 90%. Assuming anaplerotic reactions account for 30% of this dark carbon fixation (line 72), then more than half of the activity remains un-assigned to a specific process. I ask the Authors to briefly discuss or present a specific hypothesis as to which other autotrophic processes may be involved, or suggest methods that can be used to unravel the sources of this unknown dark carbon fixation activity.

Comment and action: That 30% the reviewer mentions (former line 72), refers to the total carbon incorporated into biomass (including the heteotrophic incorporation of carbon typically measured via the 3H leucine incorporation), but not 30% of the dark fixation specifically. In other words, that 30% refers to the comparison between the
amount of carbon fixed via DIC fixation (measured by dark 14C incubations) relative to the amount of carbon incorporated into biomass (measured by 3H-leucine incorporation). In any case, we agree with the reviewer it is a good idea to include some more information on it, so we have included new text suggesting potential sources of dark carbon fixation (p.5, I.165-166); it reads: "This could include aerobic anoxygenic photosynthetic bacteria (AAnPB), and oxidizers of nitrite, carbon monoxide, sulfur, etc (Hügler and Sievert, 2011)."

3. Reviewer Technical corrections Abstract: Choose one term throughout the text for consistency: "dark carbon fixation", "dark DIC fixation" or "dark CO2 fixation".

Action: Done.

4. Reviewer Line 59: text states "citation on light sensitivity" please include reference.

Action: Done.

5. Reviewer Line 114: please state actual maximum value instead of ">2.5".

Action: Done.

6. Reviewer line 122: replace "sigma-t" with "density ( $\sigma$ t)" ; after "[...] same period" add "Fig. 2)."

Action: Done.

7. Reviewer Line 123: change "(Fig 2)" to "(Fig 2C)".

Action: Done.

8. Reviewer Line 123-125: When explaining the seasonality observed at Bats for the dark to light carbon fixation ratio the Authors mention a deepening of the deep chlorophyll maximum but do not explicitly describe how this affects the calculated ratio. I ask the Authors to rephrase these sentences so that the readers can clearly understand the connexion that is currently implied in the text.
Action: We have explained this now in the text (p.4, I.132-135); it reads: "Thus, this relative decrease in chlorophyll-a (and PP) relative to the dark DIC fixation might explain the increase in the dark to light DIC fixation ratio in recent years, while also suggesting that autotrophic DIC fixation seems more sensitive to a deepening of the mixed layer than dark DIC fixation."

9. Reviewer Line 136-149: include a reference to Table 1 to guide the reader through the calcula- tions.

Action: Done.

10. Reviewer Line 138: change "[. . .] by (Yool et al., 2007)" for "[. . .] by Yool et al. (2007)".

Action: Done.

11. Reviewer Line 139-140: state the NH4+ values taken from Segura-Noguera MM et al 2014 and Lipschultz 2001 to estimate nitrification either in the text or in table 1.

Action: Done.

12. Reviewer Line 141: Dore & Karl are written twice. Correct reference.

Action: Done.

13. Reviewer Line 156: "during over the diel cycle" please correct wording.

Action: Done.

14. Reviewer Line 158-159: add a reference to Table 1.

Action: Done.

15. Reviewer Line 163: add citation for the concept that nitrification is reduced in light. Action: Done.

16. Reviewer Figure 1: keep the same range for the ratio for both plots, from 0 to 3, so

BGD
that the colour scheme is the same.

Comment: Done.

17. Reviewer Figure 2: remove names from plots (upper right corner) or write correctly (not Temp but Temperature).

Action: Done.

18. Reviewer Table 1: last column "% of dark DIC fixation from other chemolithoautotrophic and anaplerotic processes to total PP" is not explained or referred to in the text for either site. Please erase from table. Change "chemolithoautotrophy" for "chemoautotrophy".

Action: Done.

---

## Author Comment (AC3) · 3 Sep 2019

Reviewer #2

1. Reviewer: This is an interesting small paper that reviews data on dark 14C incorporation in the ocean and that postulates that it amounts to a relevant % of total primary production and that should be considered in evaluations of global primary production. I'm sympa- thetic with the author's effort as I had somehow surprisingly been puzzled by the lack of reference to dark C fixation (which it was a classic in the 80s, considered as "errors" of the Steeman-Nielsen method) I like the paper, I find the issue sensitive, and the anal- ysis is certainly worthwhile.

Comment: We honestly appreciate the positive words and support of the reviewer.

2. Reviewer: There are only a couple of points that could be discussed and that would benefit the ms. First point is stated at line 85: "dark C fixation had been attributed to the inaccuracy of the 14C method. . ." Could you expand on that? Could you tell the reader why the authors at the time thought this was an error? Why dark fixation was never considered primary production? Maybe this was due to the authors considering dark fixation as, at least in part, abiotic fixation? How do you deal with abiotic fixation in your estimates?

Comment: We had already provided an explanation of what it was meant by that in the first paragraphs of the introduction. As we explained in those paragraphs, the 14C method was developed with the aim to quantify the "photosynthetic" carbon production, so that is why they were mostly focused on what happened in the "light" incubations. That is why, is understandable, that from that point of view, the fixation that took place in the dark would be more like an error, since in general they were not considering processes that would fix DIC in the dark to be of importance. However, during the last decades we have learnt a lot about potential metabolic processes that can and do perform DIC fixation in the light. Concerning the abiotic DIC fixation, that was an issue until 1979, when Lean & Burnison (doi: 10.4319/lo.1979.24.5.0917) introduced the HCl treatment correction. They showed that when this step is performed (removal of inorganic 14C by acidification) adsorption becomes negligible. In our case, the data we used was generated by BATS and HOTS, in which this step was routinely performed.

3. Reviewer: A second point concerns to the night extrapolation of the daytime dark incorporation rates. The authors correctly identify mechanisms by which one should not assume nighttime fixation to be equal to daytime fixation (lines >160). However, I wonder how diel changes in organism activity or in water chemistry warrant that the daytime dark fixation should be above or below the night time value. Did anyone ever measure nighttime dark fixation?.

Comment: That is an interesting point, but to our knowledge no one has done that. Probably the reason for that could be that scientists in the field measuring 14C fixation were mostly interested in photosynthesis, and therefore they would do the incubations during daytime.

4. Reviewer: A third issue that could be expanded is the Table 1 increase in % dark incorporation in the 70-150 m layer. I think it was a good idea splitting the calculations by layer, but you should maybe make very clear whether this layer contains the DCM in all cases and then speculate as to why the DCM or the layer below the DCM should have a larger proportion of chemoautotrophs or anaplerotic reactions. Also, maybe the layer split could be made more clearly separating above-DCM, in-DCM and below-DCM depths.

Comment: The main reason why we decided to not only provide the integrated value for the whole layer but to also split it into two layers was because we realized that there was a clear depth-related pattern (increase) in the dark/light DIC fixation (Figure 1). We also thought about the DCM, and its potential influence. We realized that for both stations, most of the times (excluding when surface spring blooms) the DCM was at 65-75 or deeper (Fig. 2D). Based on the available sampling depths for BATS (i.e., 0,50,75,100,150 m) and ALOHA (i.e., 5, 20, 40, 65, 100, 140 and 175 m), the dark/light DIC fixation plots/dynamics (Figure 1) and on the position of the DCM (Fig. 2D) we decided to split into two layers at the sampling depth of 75 for BATS and 65 for ALOHA.

Action: We have now mentioned in the text the relative position of the DCM and how it related to the depth layers we defined (p.4, l.140-141); it reads: "The deep chlorophyll maximum (DCM) was located, most of the times (except during spring blooms), in the deeper layer (Fig. 2D)."

5. Reviewer: Finally, I'm uneasy about the 4x difference in estimations between ALOHA and BATS. I can't find any hint of the reasons for the differences, other than

different people doing the estimations. You should recognize this difference and suggest an explanation if at all possible. Can the differential oceanography of both sites play a role?.

Comment and Action: It is difficult to know exactly the reason why, since for that we would required a much more extended and deep knowledge of the physiology of autotrophic, chemotrophic and anaplerotic organisms/processes than what is nowadays available. Nevertheless we have recognized the difference as suggested by the reviewer and suggested a potential argument in that respect (p.4, l.146-151); it reads: "The reasons for these differences found between BATS and ALOHA are unknown but could be related to the contrasting nature of primary production found in these regions. In BATS, a negligible contribution from N2 fixation to N budget has been found from $\delta$15N budget exercises (Altabet, 1988) and inversion models (Wang et al., 2019). In contrast, in ALOHA, $\delta$15N budgets and inversion models estimate that 30% to 50% of export production is sustained by N2 fixation (Karl et al., 1997; Wang et al., 2019)."

6. Reviewer: Also, and about the shift of dark C fixation (or at least the proportion) occurring at BATS after 2013, I would appreciate a little bit of hypothesis-building providing a mechanistic linkage between the deepening of the mixed layer and the beneficial? effect on anaplerotic fixation (why should it be bene- fited?) or chemoautotrophy.

Action: We have explained this now in the text (p.4, l.132-135); it reads: "Thus, this relative decrease in chlorophyll-a (and PP) relative to the dark DIC fixation might explain the increase in the dark to light DIC fixation ratio in recent years, while also suggesting that autotrophic DIC fixation seems more sensitive to a deepening of the mixed layer than dark DIC fixation."

7. Reviewer: And just a tiny other comment: l. 59. Citation missing here!

Comment: Done

8. Reviewer: Good paper that should be published. My comments point to clarifications

and further insight that would, I believe, make the authors' point even stronger.

Comment: We appreciate again the contribution and support of the reviewer.

---

## Author Comment (AC4) · 3 Sep 2019

Reviewer #3

1. Reviewer: Having started working on this topic in deep-sea sediments myself, I found the manuscript very interesting, well written and worthy of publication in Biogeosciences. I have a few comments and suggested changes that I think would be good to incorporate before being finally accepted in Biogeosciences. I wish the authors all the best in revising the paper and look forward to the final publication. Please find my suggestions and comments in the attached PDF. Sincerely, Andrew K. Sweetman

Comment and action: We honestly appreciate the positive comments and the support of the reviewer. The reviewer provided his comments on a supplement pdf file. We have addressed all the comments raised by the reviewer in that supplement file.